# Validation of a Molecular Diagnostic Test for Circulating Tumor DNA by Next-Gen Sequencing

**DOI:** 10.3390/ijms242115779

**Published:** 2023-10-30

**Authors:** Sandra V. Fernandez, Yin Fei Tan, Shilpa Rao, Patricia Fittipaldi, Fathima Sheriff, Hossein Borghaei, Efrat Dotan, Jennifer S. Winn, Martin J. Edelman, Joseph Treat, Julia Judd, R. Katherine Alpaugh, Y. Lynn Wang, Jian Q. Yu, Mariusz Wasik, Don A. Baldwin

**Affiliations:** 1Department of Pathology, Fox Chase Cancer Center, Philadelphia, PA 19111, USA; yinfei.tan@fccc.edu (Y.F.T.); shilpa.rao@fccc.edu (S.R.); yuelynn.wang@fccc.edu (Y.L.W.); mariusz.wasik@fccc.edu (M.W.); 2Protocol Support Laboratory, Fox Chase Cancer Center, Philadelphia, PA 19111, USA; patricia.fittipaldi@fccc.edu (P.F.); r.alpaugh@fccc.edu (R.K.A.); 3Office of Clinical Research, Fox Chase Cancer Center, Philadelphia, PA 19111, USA; fathima.sheriff@fccc.edu; 4Department of Medical Oncology, Fox Chase Cancer Center, Philadelphia, PA 19111, USA; hossein.borghaei@fccc.edu (H.B.); efrat.dotan@fccc.edu (E.D.); jennifers.winn@fccc.edu (J.S.W.); martin.edelman@fccc.edu (M.J.E.); joseph.treat2@fccc.edu (J.T.); julia.judd@fccc.edu (J.J.); 5Department of Radiology, Fox Chase Cancer Center, Philadelphia, PA 19111, USA; michael.yu@fccc.edu

**Keywords:** ctDNA, cfDNA, clinical cancer genotyping, laboratory-developed test, assay performance assessment

## Abstract

A modified version of the PGDx elio^TM^ Plasma Resolve assay was validated as a laboratory-developed test (LDT) for clinical use in the Molecular Diagnostics Laboratory at Fox Chase Cancer Center. The test detects single nucleotide variants (SNVs) and small insertions and deletions (indels) in 33 target genes using fragmented genomic DNA extracted from plasma. The analytical performance of this assay was assessed with reference standard DNA and 29 samples from cancer patients and detected 66 SNVs and 23 indels. Using 50 ng of input DNA, the sensitivity was 95.5% to detect SNVs at 0.5% allele frequency, and the specificity was 92.3%. The sensitivity to detect indels at 1% allele frequency was 70.4%. A cutoff of 0.25% variant allele frequency (VAF) was set up for diagnostic reporting. An inter-laboratory study of concordance with an orthologous test resulted in a positive percent agreement (PPA) of 91.7%.

## 1. Introduction

Circulating tumor DNA (ctDNA) is tumor-derived fragmented DNA shed into a patient’s blood and detected in the cell-free plasma fraction. The ctDNA quantity can vary among individuals and depends on the type of tumor, location, stage, tumor burden, and response to therapy [1]. ctDNA sequences can reflect genetic heterogeneity within primary tumors or across metastatic sites that may not be observed in a tissue specimen, and plasma is easier to access serially for ongoing tumor monitoring. ctDNA assays have high sensitivity to detect single nucleotide variants (SNVs) and small insertions and deletions (indels) but have lower sensitivity for gene fusions and copy number alterations that are more difficult to detect in fragmented DNA [2]. A major limitation of liquid biopsy is sensitivity, with a risk of false negative results and the potential for false positives arising from clonal hematopoiesis of indeterminate potential (CHIP) mutations [2].

Diagnostic ctDNA applications include tumor molecular profiling, monitoring therapy response, detection of minimal residual disease (MRD), and monitoring molecular relapse (MR) [1]. For advanced cancer genotyping, ctDNA testing is appropriate when tissue is not available, is technically challenging, or prone to delay in sample acquisition. There is evidence indicating that ctDNA testing can be used in routine practice for advanced disease genotyping, with similar efficacy to tissue sequencing, to guide therapy by testing for SNVs and small indels in lung, breast, pancreatic, colorectal, gastric, hepatocellular, ovarian, urothelial, thyroid, and endometrial cancers and cholangiocarcinoma and soft tissue sarcoma [2]. The evidence base is now sufficiently strong that ctDNA has clinical utility in guiding therapy for tier I actionable variants—an approach endorsed by some specialist societies [3]. Due to its short half-life, ctDNA allows real-time monitoring of advanced cancers during therapy. Repeat liquid biopsies may allow the detection of acquired resistance variants and inform the selection of the next line of therapy. Studies monitoring cancer patients through therapy have shown that ctDNA dynamics correlate with treatment response and may identify responders earlier than clinical/radiological detection [4,5,6,7]. However, detection of MRD or MR requires assays specifically optimized for this setting that can detect low ctDNA concentrations (i.e., <0.01%) and assays intended for diagnostic cancer genotyping may not offer sufficient sensitivity [2].

Several clinical laboratories are making strong efforts to perform plasma ctDNA sequencing in-house using assays that provide an end-to-end solution from DNA library preparation to bioinformatics analysis without the need for a bioinformatics team. In the present work, PGDx elio^TM^ Plasma Resolve (EPR), a next-generation sequencing (NGS) assay, was used to develop a clinical laboratory-developed test (LDT) for detection of SNVs and short indels. After modifying the PGDx EPR data analysis workflow, the LDT was validated for tumor genotyping and monitoring therapy response in advanced solid tumors (e.g., lung, colorectal, and breast cancers). This study was designed following the College of American Pathologists guidelines for the validation of targeted NGS assays [8,9,10].

## 2. Results

### 2.1. Analytical Performance of the EPR Assay Using Reference Standards

The EPR assay is a hybrid capture ctDNA-based NGS assay targeting a 33-gene panel designed to analyze genes commonly mutated in solid tumors (Appendix A). These genes were selected by PGDx to focus on biomarkers for approved cancer therapies or relevant to established professional guidelines. The PGDx EPR assay includes reagents for NGS library preparation and a dedicated PGDx analysis server and software (version 1.7). The general performance of the EPR assay was assessed using Seraseq ctDNA Reference Material v2 containing known mutations (Appendix A) with a range of variant allele frequencies (VAF: 2%, 1%, 0.5%, 0.25%, 0.125%, and wild type) [11]. The EPR panel can potentially detect 15 SNV and 9 indels in these reference materials (Appendix A). We evaluated a range of test performance characteristics, including sensitivity, specificity, limit of detection, reproducibility, repeatability, and linearity [12]. Libraries were prepared from each reference sample using 30–60 ng input DNA with at least three replicates. A total of 76 libraries were prepared, sequenced using an Illumina NextSeq 500, and analyzed using the standard PGDx EPR algorithms for sequence alignment and variant calling. The depth of coverage (unique reads, hereafter termed “depth”) increased significantly with increasing input DNA (*p* = 0.0015), with median depths of 3841 for 30 ng, 4400 for 40 ng, 5069 for 50 ng, and 5130 for 60 ng of DNA. There were significant increases from 30 ng to 50 ng and from 30 ng to 60 ng but no significant increase from 50 ng to 60 ng (Appendix A). 

The VAFs detected by the EPR assay in these reference materials are shown in Appendix A. The sensitivity of the assay to detect SNVs and indels using different control DNA inputs is shown in Table 1. When 50 ng was used for library preparation, the EPR assay detected all SNVs at VAF 2% and 1%, and it detected 95.5%, 73.3%, and 35.5% of SNVs at VAFs of 0.5%, 0.25%, and 0.125%, respectively (Table 1). Although nine indels are potentially detected by the EPR panel, the sensitivity of the assay to detect indels was low because two indels were always filtered by the EPR software (version 1.7). The variants, APC c.4666_4667insA; p. T1556fs*3 (chr5:112175952insA) and TP53 c.263delC; p. S90fs*33 (chr17:7579420delC) were filtered as “near or within repeat region” (“RPT”) (Appendix A). The APC variant is an insertion of A in a 6 bp homopolymer A region and the TP53 variant is a deletion of C from a 5 bp homopolymer C region. In conclusion, when using 50 ng of ctDNA for library preparation, the sensitivity of the assay was 95.5% to detect SNVs at 0.5% VAF and 70.4% to detect indels at 1% VAF (Table 1).

The specificity of the assay was calculated using the wild-type reference material (Appendix A). The specificity was 92.3% when 50 ng of wild type control was used (Table 2). The positive predictive value (PPV) of the assay to detect SNVs and indels was 97.3% (Table 2). For SNV, the lower limit of detection—95% (LoD-95) of the assay was 0.5% VAF (Table 2); however, the LoD-95 was not reached for indels (Table 2). 

False positive (FP) variants were defined as pathogenic, likely pathogenic, or variants of uncertain significance present in the EPR data but not in the reference sequences of the control materials. FP variants were reported in some libraries, including those from wild-type controls (Appendix A). Of 52 FP variants, 22 has VAF < 0.25% and 30 had VAF ≥ 0.25% (Appendix A), including 24 of uncertain significance and 6 classified as pathogenic or likely pathogenic. A clinical reporting threshold of 0.25% VAF was chosen to balance increased sensitivity with reduced false positives variants. The EPR assay did not show any FP variants in the wild-type reference sample at positions corresponding to those altered in the positive control reference samples, indicating a high specificity for these regions; however, FP variants were observed outside these regions. There was a slight increase in FP variants if more DNA was used for library preparation from the wild-type control, but this did not occur with the variant positive control samples (Appendix A). 

The manufacturer’s reference allele frequencies were compared with those from the EPR assay. To assess linearity, the observed allele frequencies using 50 ng DNA (Appendix A) were plotted against the expected values and fitted by linear regression. The assay was linear for each variant, with lower limits of detection (LLOD) varying by allele (Figure 1 and Figure 2). The EPR software uses different LLOD reporting cutoffs depending on the type of variant; SNVs and indels are reported at or above a 0.3% VAF cutoff for specified variants for hotspots in the Catalogue of Somatic Mutations in Cancer (COSMIC), and 0.5% VAF at all other positions across the panel; specific variants with clinical actionability are called down to 0.1% VAF. We measured LLODs across all reference variants using different DNA inputs (Appendix A) and confirmed that 0.25% VAF LLOD is an appropriate reporting threshold for our EPR LDT (elio plasma resolve LDT) (Appendix A).

Inter-assay concordance was measured with libraries prepared by two technologists and sequenced on different days. The concordance was 100% between the replicate libraries of the 2% and 1% VAF controls and 95.8% for the replicate 0.5% VAF libraries. Inter-assay reproducibility was also assessed using replicates of clinical samples and showed high correlation (R2 = 0.9982; *p* ˂ 0.0001) (Appendix A).

### 2.2. Clinical Sample Concordance with an Orthologous Assay 

Blood samples from 29 cancer patients were used for orthogonal testing, including cases of lung cancer (17), colon cancer (5), melanoma of the skin (4), breast cancer (2), and esophageal cancer (1) (Appendix A). Most samples were from stage III or IV cancers except for RW-024 and JM-063, which were both early-stage lung adenocarcinomas (Appendix A). PGDx EPR sequencing generated an average of 54.3 million ± 7.5M (SD) read pairs per sample. The average alignment rate was 99.99% and the median depth at called variant sites was 2829 ± 1026 (SD) (Appendix A). The resulting variants were compared with data for the same samples submitted to Tempus Laboratories for genotyping by the Tempus xF ctDNA assay, a previously validated LDT. Tempus xF is a hybrid capture NGS assay that targets 105 cancer genes, of which 29 are included in the EPR panel [13]. Tempus xF does not cover four EPR panel members, BRIP1, CSF1R, POLD1, and POLE. Purified ctDNA samples were sent to Tempus and the SNVs and indels detected by the Tempus xF test were evaluated. All clinical samples harbored at least one variant detected by EPR and/or Tempus xF except for two samples, CS-031 and JM-063, which did not contain any detectable variants. CS-031 was a blood sample from a colon cancer stage IIIB patient collected post-treatment (before cycle 3, day 1), and JM-063 was a blood sample from an early stage lung cancer patient. The 81 variants identified in the 27 clinical samples are shown in Table 3. The variants BRAF c.1799T>A p.V600E; KRAS c.35G>A p.G12D; and TP53 c.818G>A p.R273H were the most common and occurred in three different patients (Table 3). Using the PGDx pipeline for analysis of the EPR data and considering only clinically relevant variants (pathogenic or likely pathogenic), the positive percent agreement (PPA) between the EPR and Tempus xF assays was 83.3% (Table 4). Unfiltered variant call format (vcf) files from the PGDx EPR were investigated for variants present in the Tempus reports but absent in the PGDx reports. As indicated in Table 3, the EPR algorithm filtered out six variants as germline (“GRM”), of which two were pathogenic or likely pathogenic (ATM c.1339C>T and BRAF c.1447A>G). Three additional variants were filtered because they were located near or within repeat regions (“RPT”), one of which was pathogenic (TP53 c.673-1G>T). Another likely pathogenic variant was filtered as complex (“CMPLX” CDH1 c.61_86delCTCTGCCAGGAGCCGGAAGCCCTGCCAinsGCT), and two likely pathogenic variants APC c.3957_3964delTGTGAGCG and APC c.4219_4234delAGTGAACCATGCAGTT with VAFs of 26% and 51%, respectively, were filtered with no reason provided in the vcf files (Table 3).

### 2.3. New Pipeline for Variant Reporting

Cancer-relevant pathogenic and likely pathogenic variants with high VAFs were filtered from the final report generated by the standard PGDx analysis algorithm; therefore, modifications were made for the LDT data analysis pipeline. Unfiltered EPR vcf files were processed to remove variants with only one read. The vcf files were then used for variant classification by Qiagen Clinical Insight (QCI) Interpret software (version 9.1.1), and: (1) variants that were pathogenic or likely pathogenic and previously filtered as germline were retained; (2) pathogenic or likely pathogenic variants with a VAF ≥ 10% and previously filtered as near or within a repeat region (“RPT”), a complex variant (“CMPLX”), or with no specified reason in the “FILTER” column of the vcf files, were retained; and (3) a clinical reporting LLOD threshold of VAF ≥ 0.25% was applied for all variant classes. Using these modified criteria, the PPA between our EPR LDT and the Tempus xF results was 91.7% for clinically relevant variants (Table 4). 

Overall, the most frequently mutated genes were TP53 (52% patients) and KRAS (26%) (Table 3). In lung cancer patients, the most frequently mutated genes were TP53 (53% patients), EGFR (23.5%), and KRAS (17.7%); in colorectal cancer, KRAS (80%), APC (80%), and TP53 (60%) (Table 3). 

There was a positive correlation for the VAF values reported by the EPR and Tempus xF assays (Figure 3).

### 2.4. Digital PCR

Of the 81 variants found in the clinical samples, 20 were selected for confirmation by digital PCR genotyping (dPCR, Table 5). dPCR confirmed high allele frequency variants filtered by the EPR algorithm but retained by the LDT pipeline in APC S1407fs*3 and BRAF K483E (Table 5). Discrepant results were observed for five variants when the three methods were compared. The variant BRAF c.793G>C, detected by EPR at 14.3% VAF but not detected by Tempus xF, was detected by dPCR at 10.57% VAF (SD = 0.35%) (Table 5), indicating that it was a true positive variant not detected by the orthologous assay. From four variants detected by EPR but not by Tempus xF, only EGFR E746_S752delinsA was also detected by dPCR (Table 5). 

### 2.5. Response to Treatment

Paired pre- and post-treatment blood samples from ten patients (three NSCLCs, six colon cancers, and one breast cancer, all stage IV, treated with any drug) were profiled using the EPR assay (Table 6). The plasma ctDNA concentrations before treatments were 7.8–683 ng; after treatment, the concentrations decreased to 10.6-37.8 ng ctDNA/mL plasma. The variants APC p. S1407fs*3 and BRAF p. K483E in patients EW-034 and MM-015, respectively, showed high VAFs (>50%) before treatment that dropped to 12% and 2.19%, respectively, after treatment, indicating that these variants were not germline as filtered by the PGDx EPR algorithm. Some patients did not show differences in the allele frequencies of the variants before and after treatment, though others showed significant decreases in the allele frequencies of some variants after treatment. The patients HN-008 and MM-015 had significant VAF decreases, indicating a possible response to treatment. Post-treatment variants were undetectable by EPR for HN-008, although dPCR detected the EGFR and TP53 variants at 0.003% and 0.086% VAF, respectively. Computed tomography (CT) scans from these patients showed overall improvements, although treatment was stopped for patient MM-015 due to toxicity. The patients EW-034 and SVO-002 also showed improvements on the CT scans after treatment and a decrease in the VAFs of mutations after treatment.

## 3. Discussion

The PGDx EPR assay was validated for NGS-based variant profiling of ctDNA as a clinical LDT at Fox Chase Cancer Center. As previously described [2], ctDNA assays have lower sensitivity for the detection of gene fusions and copy number (CN) alterations, so this LDT was initially validated only for SNVs and short indels. 

The LDT incorporates a modified version of the EPR analysis pipeline. The original algorithm removes clinically relevant variants with high VAF as potentially of germline origin. We demonstrated that some of these variants show substantially decreased VAF after treatment and were therefore somatic. Clinical reports will include high frequency variants, noting the potential for germline origin when appropriate, and providing context from matched tumor plus normal tissue sequencing when available. We also removed filters for complex variants (“CMPLX”), repeat regions (“RPT”), and unspecified rejections if the variant is clinically relevant and has VAF ≥ 10%. For VAFs < 10%, no changes were made to the pipeline other than a 0.25% VAF cutoff as the limit of detection. The LDT reporting procedure will include review and interpretation by a molecular pathologist to evaluate clinically relevant variants that pass all analysis pipeline steps and confirm the validity of the metrics for rejected variants. The PGDx positive control provided in the assay kit contains variants with relatively high VAFs; LDT guidelines recommend the inclusion of a positive control near the lower limit of detection and our protocol will include a custom control sample to meet this requirement. 

Using 50 ng of plasma DNA as the starting material, the sensitivity to detect variants at 0.5% VAF was 95.5% and 70.4% to detect indels at 1% VAF. The sensitivity of EPR for the reference indels in this study was mostly affected by post-sequencing analysis filters, which was addressed for indels with VAFs ≥ 10% classified as “RTP” or “CMPLX” by modification of the analysis pipeline as previously discussed. More generally, however, detection of indels via the short-read sequencing (here 150 bp) of highly fragmented ctDNA (also approximately 150 bp in plasma) is expected to have less sensitivity compared with SNVs because this class of variants has a broader range of potential targets. Although detection of indels less than 20 bp usually performs similarly to SNV detection, increasing the indel length decreases the sensitivity due to library representation and sequence alignment difficulties. The assessment of long indels will thus require complementary assays, such as analysis of intact genomic DNA from circulating tumor cells or tissue biopsies, combined with long-read sequencing or probe hybridization-based detection methods. 

The specificity of the assay was 92.3–93.2% using 40–50 ng plasma DNA for library preparation, and we included in these calculations as false positives any variant detected by EPR but not present in reference sequences and classified as variants of uncertain significance, pathogenic or likely pathogenic variants. We set the validated minimum input at 40 ng but prefer 50 ng for this LDT.

In Table 1, the sensitivity at 0.125% VAF appears higher (37.7%) using a 30 ng DNA input for library preparation compared with other input amounts (33.3% using 40 ng, 35.5% using 50 ng, and 33.3% using 60 ng) for SNVs, whereas for indels it appears lower (14.8%) using 30 ng input compared with other input amounts (19.4% using a 40 ng DNA input for library preparation and 18.5% using a 50 ng or 60 ng DNA input). We performed ANOVA on the replicate VAF measurements across library inputs, and the 30 ng input is significantly different (p 0.0012) from the others. This difference is due to greater deviation from the known reference VAF; that is, 30 ng input libraries report the variants at an average of 0.2% VAF, whereas the others are closer to the true 0.125% VAF. We suspect 30 ng of input DNA suffers extra variability in read representation during library construction and that this is most apparent when attempting to detect sequences at very low VAFs. ANOVA for all other reference VAF standards did not show statistical significance for differences between the input amounts, including 30 ng of DNA.

Liquid biopsies potentially identify genetic heterogeneity missed in unsampled portions of a primary tumor or in metastatic sites. During and after therapy, plasma is an easily accessible source of tumor-specific biomarkers for measuring treatment response, residual disease burden, appearance of drug resistance mutations, and tumor recurrence. The primary challenges are the low fraction of ctDNA relative to normal cell-free DNA in blood and variants that arise due to clonal hematopoiesis of indeterminant potential (CHIP). CHIP prevalence increases with aging, in both healthy and cancer patient populations, and with certain therapeutic or environmental exposures. Although CHIP variants are most often found in canonical hematopoiesis genes not covered by the EPR panel, TP53 and KRAS mutations are also commonly associated with CHIP and may confound tumor profiling. Definitive assignment of a variant’s origin as tumor, CHIP, or germline requires sequencing of a matched whole-blood or buffy coat DNA sample.

Our EPR LDT was validated for profiling advanced solid tumors such as lung, colorectal, and breast cancers, and the intended uses include as a complement to tumor/normal tissue sequencing at diagnosis, stand-alone genotyping if tissue is not available, and longitudinal monitoring of therapy responses. If tissue DNA is not available or its sequencing would require long turnaround times, the ESMO recommends liquid biopsies for genotyping aggressive tumor types where time to result is clinically important, such as in advanced non-small cell lung cancer (NSCLC) [2]. The EPR panel targets genes that are appropriate for meeting this and other guidelines. The NCCN Guidelines for NSCLC (version 7.2021) recommend molecular testing at the time of diagnosis with repeat biopsy or plasma testing to enable identification of genomic alterations in EGFR, ALK, ROS, BRAF, MET, and RET to guide the use of FDA-approved targeted therapies in the first-line advanced disease setting [2]. In breast cancer, identification of PIK3CA mutations informs treatment with the PI3Kα-specific inhibitor alpelisib in combination with fulvestrant as a second-line therapy for advanced disease [14,15]. Accordingly, the NCCN Guidelines for invasive breast cancer (version 1.2022) recommend assessment for PIK3CA mutations using tumor tissue or ctDNA testing, with reflex tumor testing if the ctDNA results are negative [2]. For most other tumors, the NCCN Guidelines do not directly address ctDNA testing but acknowledge that relevant genomic alterations may be identified by evaluating plasma ctDNA [16]. 

A recent study found that ctDNA can serve as an adjunct to CT scans for MRD monitoring in resected NSCLC [14] and suggests that longitudinal ctDNA analysis could help clarify equivocal radiological diagnosis and be a useful complement to routine clinical imaging [14]. CT scans showed some correlation with changes in biomarker VAF for the clinical cases in this study, but extended longitudinal testing will be necessary to confirm such a correlation. 

Other NGS panels are available for ctDNA profiling or have been successfully deployed as clinical ctDNA LDTs. Roche’s Avenio ctDNA Expanded assay is a 77-gene panel covering 567 known hotspot variant regions selected for different solid tumors, mainly NSCLC and colorectal cancer, that has been validated for clinical use by several laboratories [13,14,15,16]. The MSK-ACCESS is another ctDNA clinical test developed and used by Memorial Sloan Kettering that can detect genomic alterations (SNV, indels, CN, and structural variants or SVs) in select regions of 129 key cancer-associated genes; the test was approved for diagnostic use by the New York State Department of Health in 2019 [17]. Other commercially available tests include the Illumina TruSight Oncology 170 ctDNA and TruSight Oncology 500 ctDNA kits [14,18,19]. The Sequencing Quality Control Phase 2 (SEQC2) consortium assessed the analytical performance of five ctDNA assays including Illumina TruSight Oncology 170, Avenio ctDNA Expanded, and panels from Integrated DNA Technologies, Burning Rock Dx, and Thermo Fisher Scientific. The results demonstrated good performance across platforms, with consistent detection of variants above 0.5% VAF and high sensitivity, precision, and reproducibility for all five assays [18,20].

In conclusion, the EPR LDT validation demonstrated the good performance of the assay for detection of SNVs and indels after some modification to the reporting pipeline. We will extend our efforts on the validation of larger panels that can be used to monitor a wider spectrum of other cancers. Increased sensitivity may ultimately be required for ctDNA to be fully informative regarding residual tumor burden, accompanied by sequencing of therapeutic targets to detect resistant subclones that may emerge during treatment.

## 4. Materials and Methods

### 4.1. Samples, Storage, and ctDNA Isolation 

The ctDNA reference standards were Seraseq ctDNA Reference Material v2 (SeraCare Life Science, Milford, MA, USA). These consist of DNA from cell line GM24385 as a background wild-type diluent, plus constructs containing several variants. Each control is available as isolated ctDNA (10 ng/µL; total mass = 250 ng) at several VAFs: 2% (Cat# 0710–0139), 1% (Cat# 0710–0140), 0.5% (Cat# 0710–0141), 0.13% (Cat# 0710–0143), and wild-type (Cat# 0710–0144) or in 5 mL plasma aliquots with 25 ng/mL DNA. Allele frequencies determined via droplet digital PCR and provided by the manufacture were compared with the EPR quantitative data. Libraries were prepared with these controls using 30 ng, 40 ng, 50 ng, and 60 ng as input ctDNA. These reference samples contain 40 clinically relevant variants (SNVs, indels, and structural variants or SVs) in 28 genes, all at the same specified VAF (Appendix A). 

An additional 29 blood samples from cancer patients were used to assess the clinical concordance between the EPR and Tempus xF assays. The samples were collected through the Fox Chase Cancer Center protocol IRB 19-9030 and patients signed informed consent forms. To study ctDNA in response to treatment, 10 patients were enrolled (6 colon cancer, 3 NSCLC, and 1 breast cancer) to have blood samples drawn before starting a new drug treatment (pre-treatment) and post-treatment. Two tubes of blood were collected from each patient in Cell-free DNA BCT collection tubes (Streck, La Vista, NE, USA) and plasma was isolated within 48 h. The whole blood was centrifuged at 2000× *g* for 10 min at room temperature. Isolated plasma was collected and centrifuged a second time at 16,000× *g* at 4 °C for 10 min and stored at −80 °C until DNA extraction. The buffy coat was also collected and stored at −80 °C for future analysis of germline and CHIP variants. DNA was isolated using the Qiagen QIAamp MinElute ccfDNA Midi kit (Qiagen) according to manufacturer’s instructions and quantified using the Qubit dsDNA High-Sensitivity kit (Thermo Fisher Scientific, Waltham, MA, USA). Sample quality was confirmed using the High-Sensitivity DNA kit on the Agilent 2100 Bioanalyzer.

### 4.2. EPR Library Construction and Sequencing

The EPR assay (PGDx Inc./Labcorp Holdings, Baltimore, MD, USA) includes reagents for NGS library preparation and a dedicated PGDx analysis server and software. The targeted hybrid capture panel identifies variants in the full coding regions of 33 genes covering 117,602 nucleotides. The panel interrogates SNVs and short indels, amplifications in 8 genes, 4 translocations, and microsatellite instability status (Appendix A). The PGDx EPR positive control consists of 3 SNVs, 2 indels at ~5% VAF, 2 fusions, and 2 copy number (CN) variants. Library construction was performed following the PGDx protocol. Briefly, ctDNA was mixed with reagents for end repair, dA addition, and adaptor ligation. A set of 16 indexed adaptors with variable 6-bp DNA barcodes was ligated randomly onto both ends of each input duplex ctDNA fragment. Ligated sequencing libraries were PCR amplified with a universal PCR primer and an indexed PCR primer. Capture probe hybridization was performed following the protocol. Library concentration was assessed using the Qubit dsDNA High Sensitivity Assay followed by the Bioanalyzer High Sensitivity DNA kit on Bioanalyzer to determine the size of each library. Libraries were normalized to 1.05 nmol/L. Eight libraries, including one prepared with the PGDx positive control, were pooled together, loaded in a flow cell, and sequenced on a NextSeq 500 (Illumina, San Diego, CA) using the high-output sequencing kit and 150 cycle, dual-indexed, paired-end reads.

### 4.3. Bioinformatics Pipeline

Sequencing data were processed on the PGDx server with analysis software version 1.7. The EPR pipeline performs read deconvolution and alignment, variant calling, and quality control filtering [21]. The pipeline was run in-house on the server and all patient data were stored locally behind the laboratory firewall. Two default reports were automatically generated as output from the server for each sample: (1) an unfiltered variant report (unfiltered vcf) and (2) a filtered variant report (vcf) applying standard PGDx EPR criteria. SNVs and indels are reported at or above a 0.3% VAF cutoff for specified variants for hotspots in COSMIC, and 0.5% VAF at all other positions across the panel; specific variants with clinical actionability are called down to 0.1% VAF. The filtering rules in EPR do not allow reporting of insertions or deletions in homopolymers ≥ 4 bp in length, as insertions and deletions from homopolymer regions tend to be lower confidence calls; these variants are filtered as “RPT” (near or within repeat regions). QCI Interpret (Qiagen) was used for variant interpretation and classification.

### 4.4. EPR Technical Performance Assessment

The EPR assay can potentially detect 15 substitutions and 9 indels in the reference materials (Appendix A). Each EPR test was run with a minimum of three replicates at 30 ng, 40 ng, 50 ng, and 60 ng of ctDNA input. Sensitivity, specificity, PPV, and LoD-95% were determined using these reference materials. Sensitivity is defined as the proportion of known variants that are correctly classified as positive, that is, the likelihood that the assay will detect a variant if it is present. The sensitivity was calculated by dividing true positive (TP) calls by the sum of TP and false negative (FN) calls. The positive predictive value (PPV) was calculated by dividing TP calls by the sum of TP and false positive (FP) calls. False positive (FP) variants were defined as pathogenic, likely pathogenic, or variants of uncertain significance present in EPR data but not in reference sequences of the control materials. The specificity was calculated using the wild-type (WT) control and calculated by the number of known true negative (TN) variants divided by the number of true negative (TN) plus FP variants identified by the assay (specificity = TN/(TN + FP)), and it is defined as the ability of a test to detect only the targets. The 95% limit of detection (LoD-95) is defined as the minor allele fraction at which 95% of samples are detected [9]. Different mutation types, such as SNVs and indels, may have different LoD-95s, and therefore, the LoD-95 was determined for each variant type.

### 4.5. EPR Clinical Concordance Assessment

To establish the PPA, the EPR results from 29 clinical samples were compared with the results of an orthogonal reference test, Tempus xF (Tempus Laboratories, Chicago, IL, USA). The Tempus xF assay is a hybrid capture NGS ctDNA assay covering 105 genes to assess SNVs, indels, CN variants, and translocations. The assay typically uses 30 to 50 ng of input DNA and spans clinically relevant coding exons for 35 genes and recurrent hotspot mutations in 70 genes. ctDNA aliquots (~40 ng in 50 µL) isolated from patient plasma were sent to Tempus Laboratories and compared with the EPR results. Variants that were off-target or below the LLOD (Tempus cutoff 0.3% VAF; EPR LDT cutoff 0.25% VAF) were excluded from the analysis. PPA is used when the new test (EPR) is compared with a non-reference standard (Tempus xF) [8].

### 4.6. Precision: Repeatability and Reproducibility

The repeatability of the EPR assay was determined by comparing variant detection in the following reference material libraries: YT-LIB22 vs. YT-LIB23 for variants at 1% VAF; libraries YT-LIB 21 vs. YT-LIB 19 for variants at 0.5% VAF; and Lib209 vs. Lib210 for variants at 0.25% VAF (Appendix A). For inter-operator reproducibility, libraries were prepared by different operators and sequenced on separate runs. Inter-operator reproducibility was studied similarly and assessed: (1) Lib119 vs. YF- VAF2, 2%VAF; (2) Lib121 vs. Lib. YT-LIB22, 1% VAF; and (3) Lib123 vs. Yinf2, 0.5% VAF (Appendix A). To determine inter-run reproducibility, clinical samples were used and the same libraries were run twice on separate flow cells: libraries from Run #8 vs. libraries from Run #9 (Appendix A). Data were compared from the following libraries: (1) Lib151-Run #8 vs. Lib151b-Run #9 (SVO-002 post-treatment colon cancer); (2) Lib 152- Run #8 vs. Lib 152b-Run #9 (KRS-014 colon pre-treatment); (3) Lib 153- Run #8) vs. Lib 153b-Run #9 (KRS-014 colon post-treatment); (4) Lib 154-Run #8 vs. Lib 154b-Run #9 (MM-015 colon pre-treatment); (5) Lib 155- Run #8 vs. Lib 155b- Run #9 (MM-015 post-treatment); (6) Lib 156-Run #8 vs. Lib 156b- Run #9 (EW-034 colon pre-treatment); and (7) Lib 157- Run #8 vs. Lib 157b- Run #9 (EW-034 colon post-treatment) (Appendix A).

### 4.7. Digital PCR

Probes and primers were designed using Integrated DNA Technologies (IDT) PrimerQuest software (version 2.2.3) and were synthesized by IDT (Coralville, IA, USA). The wild-type probes were labelled with HEX dye and the mutant probes were labelled with FAM dye. The digital PCR was performed in the QIAcuity One instrument (Qiagen). The QIAcuity Probe PCR kit (Qiagen Cat# 250102) was used to set up the 40 µL reaction, 20 ng of ctDNA was used as input, and the reaction was carried out in the QIAcuity 26k Nanoplate (Qiagen, Cat# 250001). The cycling conditions were: initial denaturing at 95 °C for 3 min; then 40 cycles at 95 °C, 20 s; 60 °C, 20 s; and 68 °C, 30 s. Imaging was set for green (FAM dye) and yellow (HEX dye) channels. The QIAcuity software suite 2.2.0.26 (Qiagen) was used for data analysis. For patient samples with a limited amount of ctDNA, the NEB Ultra II kit (New England Biolabs, Ipswich, MA, USA #E7645) was used to amplify ctDNA. A total of 20 ng ctDNA was used as input and 8 PCR cycles were used to produce around 2500 ng of amplified ctDNA. An amount of 100 ng amplified DNA was used per dPCR reaction. The primer/probe sequences are listed in Appendix A.

## 5. Conclusions

This study describes the validation of the EPR assay for use as a clinical molecular diagnostic test. We concluded that 40–50 ng of starting material should be used for library preparation. We modified the PGDx pipeline for analysis since pathogenic and likely pathogenic variants of high VAF from somatic origin were removed by the standard algorithm. Pathogenic or likely pathogenic variants with a VAF ≥ 10%, which were filtered as near or within a repeat region, and complex variants, are retained as reportable variants. Variants filtered by the PGDx pipeline as germline are also retained. We established a cutoff of 0.25% VAF for the diagnostic reporting of variants.

## Figures and Tables

**Figure 1 ijms-24-15779-f001:**
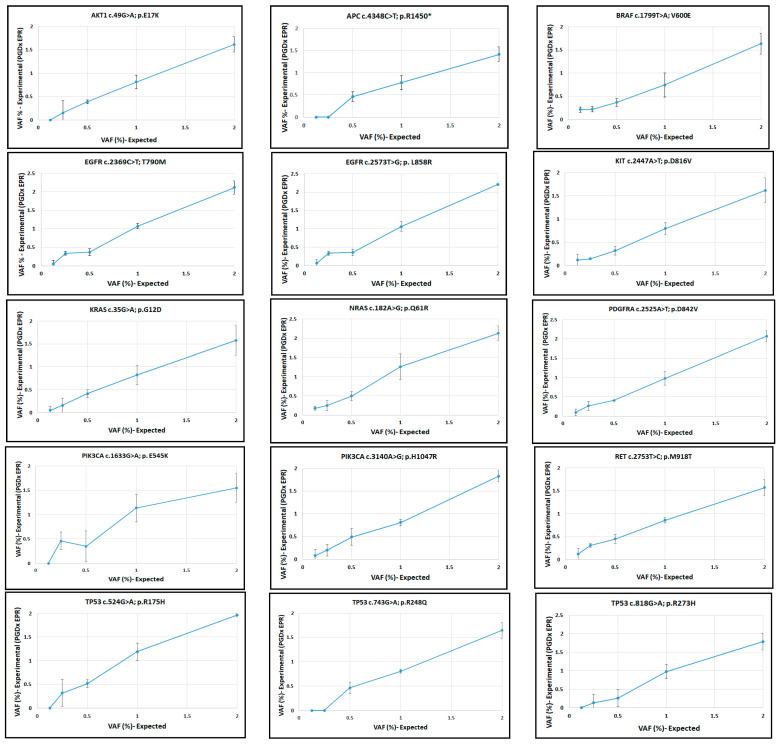
Linearity of the EPR assay for SNVs in the reference samples. Expected VAFs vs. experimental VAFs (EPR assay) are plotted.

**Figure 2 ijms-24-15779-f002:**
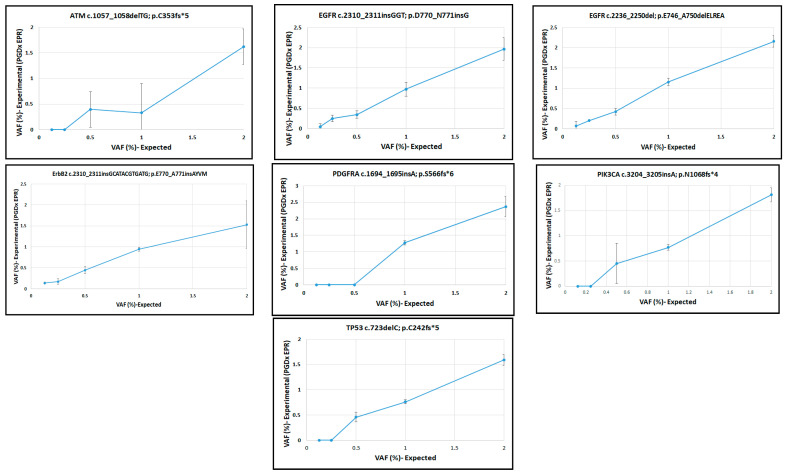
Linearity of the EPR assay for indels in reference samples. Expected VAFs vs. experimental VAFs (EPR assay).

**Figure 3 ijms-24-15779-f003:**
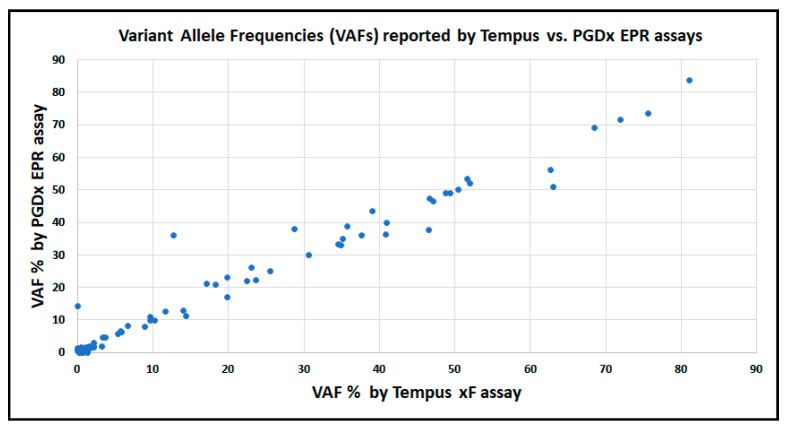
Correlation between VAFs of variants reported by Tempus and/or EPR. VAFs of all variants are shown. Equation: Y = 0.9626 X + 1.043; R2 = 0.9676; *p* < 0.0001. The variant detected by EPR at 14.3% VAF but not detected by Tempus xF was confirmed to be present by digital PCR (10.57%; SD = 0.35%).

**Table 1 ijms-24-15779-t001:** Sensitivity of the EPR assay to detect SNVs and indels in reference material. Different DNA quantities (30 ng, 40 ng, 50 ng, and 60 ng) of the reference controls were used for library preparation and the sensitivity of the assay was calculated. VAF, variant allele frequency; SNVs, single nucleotide variants; indels, small insertions and deletions.

Variant Type	DNA Quantity	At 0.125% VAF % (Detected/Total)	At 0.25% VAF % (Detected/Total)	At 0.5% VAF % (Detected/Total)	At 1% VAF % (Detected/Total)	At 2% VAF % (Detected/Total)
15 SNV	30 ng	37.7% (17/45)	64.4% (29/45)	91.1% (41/45)	100% (45/45)	97.8% (44/45)
	40 ng	33.3% (20/60)	77.7% (35/45)	93.3% (70/75)	100% (45/45)	100% (45/45)
	50 ng	35.5% (16/45)	73.3% (33/45)	95.5% (43/45)	100% (45/45)	100% (45/45)
	60 ng	33.3% (15/45)	68.9% (31/45)	93.3% (42/45)	100% (45/45)	100% (45/45)
9 indels	30 ng	14.8% (4/27)	37% (10/27)	66.7% (18/27)	74% (20/27)	77.8% (21/27)
	40 ng	19.4% (7/36)	37% (10/27)	46.7% (21/45)	77.8% (21/27)	77.8% (21/27)
	50 ng	18.5% (5/27)	33.3% (9/27)	59.3% (16/27)	70.4% (19/27)	77.8% (21/27)
	60 ng	18.5% (5/27)	33.3% (9/27)	59.3% (16/27)	70.4% (19/27)	77.8% (21/27)

**Table 2 ijms-24-15779-t002:** Specificity, positive predictive value (PPV), and lower limit of detection—95% (LoD-95). Specificity was calculated using the wild-type control. LoD-95, minor VAF at which 95% of samples would pass the detection criteria. TP, true positives; FP, false positives; TN, true negatives; SNV, single nucleotide variants.

	Seraseq ctDNA Reference Material v.2 (ng) for Library Preparation			
	Using 30 ng	Using 40 ng	Using 50 ng	Using 60 ng
PPV = TP/(TP+FP)	96.5% [249/(249+9)]	97.4% [300/(300+8)]	97.3% [252/(252+7)]	98.4% [249/(249+4)]
LoD-95 for SNVs	1% VAF	1% VAF	0.5% VAF	0.5% VAF
Specificity = TN/(TN + FP)	93.5% [72/(72+5)]	93.2% [96/(96+7)]	92.3% [72/(72+6)]	92.3% [72/(72+6)]

**Table 3 ijms-24-15779-t003:** Variants detected in clinical samples.

Gene	Variant (DNA)	Variant (Protein)	Tempus Assay VAF%	EPR Assay VAF%	Classification	Type of Tumor	Patient ID#
AKT1	49G>A	E17K	39.1	43.35	Path.	Breast	MF-053
ALK	1108G>A	E370K	35.7	38.88	Unc. Sig.	Colon	EW-034
ALK	3599C>T	A1200V	40.8	36.16	Unc. Sig.	Mel.	SH-041
ALK	3885G>A	W1295*	3.7	4.55	Unc. Sig.	Lung	JS-040
ALK	3931G>C	D1311H	3.3	4.66	Unc. Sig.	Lung	JS-040
APC	734C>A	S245*	14	12.85	Path.	Colon	HM-059
APC	4349G>A	R1450Q	48.8	0 (48.9 ^G^)	Unc. Sig.	Lung	GP-051
ARID1A	947A>T	Y316F	0.6	0	Unc. Sig.	Lung	DK-055
ARID1A	1978G>A	G660R	1.7	1.32	Unc. Sig.	Lung	CK-027
ARID1A	5965C>T	R1989*	0.3	0	Path.	Lung	AS-012
ATM	1339C>T	R447*	51.6	0 (53.4 ^G^)	Path.	Lung	RC-055
ATM	6188G>A	G2063E	17.1	21	Lik. Path.	Colon	AD-050
ATM	8165T>G	L2722R	5.4	5.54	Unc. Sig.	Lung	JR-010
ATM	9023G>A	R3008H	18.3	20.81	Path.	Colon	AD-050
ATM	9101T>G	L3034W	50.4	0 (50 ^G^)	Unc. Sig.	Lung	LTW-047
BRAF	793G>C	G265R	0*	14.3	Lik. Path.	Lung	LTW-047
BRAF	1447A>G	K483E	68.5	0 (69 ^G^)	Lik. Path.	Colon	MM-015
BRAF	1799T>A	V600E	0.6	0.69	Path.	Mel.	LH-070
BRAF	1799T>A	V600E	1.6	1.94	Path.	Mel.	JB-086
BRAF	1799T>A	V600E	19.8	17.07	Path.	Mel.	SH-041
BRCA1	3599A>C	Q1200P	0.6	0	Unc. Sig.	Lung	CL-080
BRCA2	7903G>A	E2635K	10.2	9.83	Unc. Sig.	Colon	HM-059
BRCA2	3568C>T	R1190W	52	52.11	Benign	Lung	JS-040
CCND1	835G>T	E279*	1.3	0	Unc. Sig.	Lung	AS-012
CDH1	817G>A	E273K	0.3	0	Unc. Sig.	Lung	DK-055
CDH1	2199G>T	R733S	0.3	0	Unc. Sig.	Lung	LTW-047
EGFR	2255C>T	S752F	0	1.16	Lik. Path.	Lung	AS-012
EGFR	2918G>A	R973Q	0.9	1.07	Unc. Sig.	Lung	JR-010
EGFR	3055C>T	P1019S	0.8	0.56	Unc. Sig.	Lung	RC-055
KIT	497C>G	P166R	0.6	0.78	Unc. Sig.	Lung	JK-049
KIT	1588G>A	V5301	0.8	0	Unc. Sig.	Lung	PC-075
KRAS	34G>T	G12C	0.5	1.62	Path.	Colon	EW-034
KRAS	35G>A	G12D	9.6	9.88	Path.	Lung	MS-072
KRAS	35G>A	G12D	22.5	21.85	Path.	Colon	HM-059
KRAS	35G>A	G12D	34.5	33.28	Path.	Colon	AD-050
KRAS	35G>T	G12V	14.4	11.13	Path.	Lung	CL-080
KRAS	35G>C	G12A	25.5	25.08	Path.	Lung	JS-040
KRAS	437C>T	A146V	35.1	34.9	Path.	Colon	MM-015
MET	3637C>A	L1213I	8.9	7.9	Lik. Path.	Lung	LD-030
MET	3937T>A	Y1331N	6.7	8.25	Unc. Sig.	Lung	MS-072
MYC	138C>G	F46L	23.6	22.21	Unc. Sig.	Breast	CM-029
MYC	1307G>A	R436Q	49.4	0 (49 ^G^)	Unc. Sig.	Esoph.	AG-042
NTRK1	2350C>G	L784V	0.6	0	Unc. Sig.	Lung	LTW-047
PIK3CA	1624G>A	E542K	0	0.31	Path.	Colon	HM-059
PIK3CA	1624G>A	E542K	0.3	** **0.13** **	Path.	Lung	JK-049
PIK3CA	1624G>A	E542K	81	83.73	Path.	Breast	CM-029
PIK3CA	1633G>A	E545K	0.3	0.26	Path.	Colon	HM-059
PIK3CA	1634A>G	E545G	0.3	0.16	Path.	Colon	HM-059
PIK3CA	2176G>A	E726K	0	0.32	Path.	Lung	AS-012
RET	1078C>T	R360W	0	0.9	Unc. Sig.	Mel.	JB-086
ROS1	5209G>T	E1737*	1.5	1.4	Unc. Sig.	Breast	MF-053
ROS1	6286C>T	R2096W	30.7	0 (29.8 ^R^)	Unc. Sig.	Lung	DK-055
TP53	313G>T	G105C	19.9	23.04	Path.	Lung	LD-030
TP53	469G>T	V157F	2.2	1.75	Path.	Lung	PC-075
TP53	641A>G	H214R	0.3	0.32	Path.	Esoph.	AG-042
TP53	659A>G	Y220C	2.2	2.89	Path.	Lung	JK-049
TP53	673-1G>T	Splicing region	37.6	0 (36 ^R^)	Path.	Breast	CM-029
TP53	715A>G	N239D	34.9	33.03	Path.	Lung	HN-008
TP53	733G>T	G245C	47.1	46.55	Path.	Lung	DK-055
TP53	743G>A	R248Q	5.7	6.56	Path.	Lung	JS-040
TP53	773A>G	E258G	1.1	1.43	Path.	Lung	AS-012
TP53	818G>A	R273H	0.4	0.39	Path.	Lung	LD-030
TP53	818G>A	R273H	0.8	0.6	Path.	Lung	JK-049
TP53	818G>A	R273H	46.7	47.39	Path.	Colon	EW-034
TP53	524G>A	R175H	71.9	71.44	Path.	Colon	MM-015
TP53	856G>A	E286K	5.9	6.18	Path.	Lung	CL-080
APC	3460_3462delGAA	E1154del	46.5	37.54	Lik.ben.	Lung	DH-058
APC	3956delC	P1319Lfs*2	11.7	12.67	Path.	Colon	HM-059
APC	3957_3964delTGTGAGCG	V1320fs*9	23	0 (26 ^Φ^)	Lik. Path.	Colon	AD-050
APC	4219_4234 delAGTGAAC…	S1407fs*3	63	0 (51 ^Φ^)	Lik. Path.	Colon	EW-034
APC	4326delT	P1443Lfs*30	75.6	73.48	Path.	Colon	MM-015
ARID1A	123_128delGGC	A42_A43del	62.7	0 (56 ^G^)	Unc. Sig.	Lung	GP-051
BRAF	1798_ 1799 delGTinsAA	V600K	2.2	1.66	Path.	Mel.	MI-043
BRCA2	6284delC	S2095Y fs*24	3.2	1.78	Lik. Path.	Lung	GP-051
CDH1	61_86delCTCTGCCAGGAGCCGGAGCCCTGCCAinsGCT	L21fs	12.7	0 (36 ^C^)	Lik. Path.	Breast	MF-053
EGFR	2235_2249delGGAATTA…	E746_A750del	28.8	37.8	Path.	Lung	HN-008
EGFR	2237_2254delAATTAAGAGAAGCAACAT	E746_S752delinsA	0	1.09	Path.	Lung	AS-012
EGFR	2237_2255delAATTAAGAGAAGCAACATinsT	E746_S752delinsV	1.3	1.18	Path.	Lung	AS-012
ErBB2	2230_2231 del GA	N745Cfs*128	41	0 (39.8 ^R^)	Unc. Sig.	Colon	AD-050
TP53	773_782delAAGACTCCAG	E258Vfs*84	0.5	1.01	Path.	Colon	HM-059
TP53	780delC	S261Vfs*84	9.6	10.92	Path.	Lung	RW-024

Variants detected in clinical samples. Variants detected by the Tempus xF and/or EPR assays. Variants in EPR unfiltered vcf files but filtered by the software are indicated in red: ^G^, germline; ^C^, complex; ^R^, in or near repeat region; ^Φ^, no reason provided in the vcf files. From those variants, pathogenic or likely pathogenic variants are highlighted in yellow. The Tempus xF false negative is indicated as 0*. In blue, VAFs reported by the EPR assay with values lower than the 0.25% cutoff. Lik. Path, Likely Pathogenic; Lik. ben., likely benign; Path., Pathogenic; Unc. Sig., variant of uncertain significance; Esoph., esophageal adenocarcinoma; Mel., melanoma.

**Table 4 ijms-24-15779-t004:** Positive percent agreement (PPA) with Tempus xF.

	PGDx EPR Algorithm for Variant Reporting	New Pipeline for Variant Reporting- EPR LDT
PPA considering clinically significant variants (pathogenic and likely pathogenic)	83.3% (40/(40 + 8))	91.7% (44/(44 + 4))

**Table 5 ijms-24-15779-t005:** VAFs by EPR, Tempus xF, and digital PCR. Tempus xF uses 0.3% VAF as cutoff. In red, variants with discrepant results by Tempus xF, PGDx EPR and dPCR. G, variant in EPR unfiltered vcf file but filtered as germline by the software; Φ, variant in EPR unfiltered vcf file but filtered by the software and no reason was provided in the “Filter” column of vcf file.

Gene	Variant (DNA)	Variant (Protein)	Tempus xF VAF %	PGDx EPR VAF %	dPCR (%VAF ± SD)	Patient ID#
ALK	1108G>A	E370K	35.7	38.88	38.78 ± 0.59	EW-034
ALK	3599C>T	A1200V	40.8	36.16	40.08 ± 3.4	SH-041
APC	4219_4234 del AGTGAAC…	S1407fs*3	63	0 (51 ^Φ^)	69.4 ± 0.8	EW-034
APC	4326delT	P1443Lfs*30	75.6	73.48	76.62 ± 0.13	MM-015
BRAF	793G>C	G265R	0	14.3	10.57 ± 0.35	LTW-047
BRAF	1447A>G	K483E	68.5	0 (69 ^G^)	70.11 ± 1.11	MM-015
BRAF	1799T>A	V600E	19.8	17.07	14.5 ± 3.8	SH-041
BRAF	1798_ 1799 delGTinsAA	V600K	2.2	1.66	1.74 ± 0.11	MI-043
EGFR	2255C>T	S752F	0	1.16	0	AS-012
EGFR	2237_2254delAATTAAGAGAAGCAACAT	E746_S752delinsA	0	1.09	2.083 ± 0.23	AS-012
EGFR	2235_2249delGGAATTA..	E746_A750del	28.8	37.8	41.61 ± 0.59	HN-008
KRAS	34G>T	G12C	0.5	1.62	0.916 ± 0.19	EW-034
KRAS	35G>C	G12A	25.5	25.08	13.25 ± 1.53	JS-040
KRAS	437C>T	A146V	35.1	34.9	41.27 ± 1.73	MM-015
PIK3CA	1624G>A	E542K	0	0.31	0.08 ± 0.03	HM-059
PIK3CA	2176G>A	E726K	0	0.32	0.11 ± 0.02	AS-012
TP53	524G>A	R175H	71.9	71.44	73.16 ± 2.81	MM-015
TP53	715A>G	N239D	34.9	33.03	36.19 ± 0.91	HN-008
TP53	733G>T	G245C	47.1	46.55	42.42 ± 0.75	DK-055
TP53	818G>A	R273H	46.7	47.39	47.21 ± 0.63	EW-034

**Table 6 ijms-24-15779-t006:** Variants detected using the EPR assay in ctDNA before and after treatment and CT scan results.

Cancer Type	Patient ID#	Variant	VAF Pre-Treat.	VAF Post-Treat.	CT Scan	
NSCLC Stage IV	CH-067	TP53 G245V (exon 7)- Path.	Month 0 0.95%	Month 1 0.32%	Compared Months 1 vs. -3: Lung and subcimal better **Overall: better** (Month 11 under this treatment continue responding)	Month 11- Alive
NSCLC Stage IV	HN-008	EGFR E746_750del (exon19)-Path TP53 N239D (exon 7)- Path.	Month 0 38% 33%	Month 3 (Osimertinib) Not detected ^1^ Not detected ^2^	Compared Months 3 vs. 0: Improvement of liver and bone metastases **Overall: better** (Month 24: progression)	Month 33-Expired (new brain met.)
NSCLC Stage IV	LTW-047	BRAF G265R (exon 6)- Lik. Path.	Month 0 14.3%	Month 3 22.57%	Compared Months 4 vs. -1: Lung lesion: stable **Overall: same** (Month 17: progression)	Month 23- Alive
Colon Stage IV	RA-027	KRAS A146T (exon 4)- Path. PIK3CA E545K (exon 10)-Path. TP53 R282W (exon 8)- Path. RAF1 R191I (exon 5)- Unc. sig.	Month 0 4.58% 5.10% 5.17% 6.24%	Month 1 5.33% 7.18% 7.61% 7.92%	Compared Months-2 vs. 1: New hepatic lesions and portacaval lymph nodes **Overall: worst**	Month 15- Expired
Colon Stage IV	EW-034	KRAS G12C (exon 2)- Path. TP53 R273H (exon 8)- Path. APC S1407fs*3 (exon16)-Lik.Path.	Month 0 1.62% 47.39% 51% ^G^	Month 1 5.52% 4.47% 12.60%	Compared Months 3 vs. 0: Decrease in hepatic tumor burden **Overall: better** (Month 9: progression)	Month 22- Expired
Colon Stage IV	MM-015	APC P1443Lfs*30 (exon 16)-Path. BRAF K483E (exon 12)-Lik.Path. KRAS A146V (exon 4)- Path. TP53 R275H (exon 5)- Path.	Month 0 73.48% 69% ^G^ 39.4% 71.44%	Month 1 0.94% 2.19% 1.99% 0.83%	Compared Months 4 vs. -1: Decreasing hepatic and adrenal metastases. **Overall: better** (Month 3: treatment stopped due toxicity)	Month 8- Expired
Colon Stage IV	MB-051	KRAS G12S (exon 2)- Path. TP53 R273H (exon 8)- Path. APC P1319fs*2 (exon 16)- Path.	Month 0 10.82% 11% 14%	Month 1 2.01% 1.86% 2.68%	Compared Months 3 vs. 0: Liver and lung metastases did not show change **Overall: same** (Month 7: progression)	Month 9- Expired
Colon Stage IV	SVO-002	APC Y935* (exon 16)- Path. APC P1440fs*33 (exon 16)- Path. KRAS A146T (exon 4)- Path. TP53 E51* (exon 4)- Path.	Month 0 3% 3.72% 7.65% 5.85% ^R^	Month 2 0.54% 0.31% 1.65% 1.63% ^R^	Compared Months 3 vs. 0: Decreasing hepatic met. **Overall: better** (Month 12: progression)	Month 34- Alive (stable disease)
Colon Stage IV	KRS-014	APC R564* (exon 14)- Path. NRAS Q61K (exon 3)- Path. TP53 D259Y (exon 7)- Path.	Month 0 30.09% 0.28% 19.28%	Month 2 4.44% Not detected 3.25%	Compared Months 3 vs. -1: Liver metastases worst and new lung and brain met. **Overall: worst** (Month 6: progression)	Month 15- Expired
Breast Stage IV	CA-062	PIK3CA I69N (exon 2)- Unc. Sig. PIK3CA H1047R (exon 21)- Path.	Month 0 29.34% 27.46%	Month 2 20.67% 22.23%	Compared Months 2 vs. 0: Liver met. worse; lung and bone met. stable **Overall: worst** (Month 3: progression)	Month 13- Expired

Variants were studied via the EPR assay in plasma ctDNA before and after treatment (with any drug). CT scans were evaluated by comparing them before and after treatment. ^1^ By dPCR, VAF= 0.003% (StD = 0.006%). ^2^ By dPCR, VAF = 0.086% (StD = 0.013%). Path., pathogenic; Lik. Path., likely pathogenic; Unc. Sig, variant of uncertain significance; ^G^, variant in the unfilter vcf file previously filtered by PGDx software as germline; ^R^, variant in the unfiltered vcf file previously filtered by the software as “RPT” (near or within a repeat region); met., metastases. Month -3 means that CT scan was performed 3 months before month 0.

## Data Availability

All data generated or analyzed during this study are included in this article.

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
