# Peer review of "Validation of a Molecular Diagnostic Test for Circulating Tumor DNA by Next-Gen Sequencing"

_ijms, 2023, doi:10.3390/ijms242115779_

Round 1
Reviewer 1 Report
Comments and Suggestions for Authors
Author Response
Reviewer #1
Authors compiled a manuscript entitled “Validation of a molecular diagnostic test for circulating tumor DNA by next-gen sequencing” which is the topic of choice in today’s research. If we are able to validate the ctDNA detection, it would have a great importance in early cancer detection as well as save time and cost of the treatment. Authors compiled the manuscript very nicely and can be considered for publication with addressing the minor suggestions as mentioned below.”
Response to reviewer#1: Thank you very much for your time to review our manuscript. We appreciate your feedback. All your comments and suggestions were included in the re-submitted version of our manuscript and are highlighted in blue.
- Authors enrolled only 29 patients in the study. Is this number sufficient to draw a conclusion for sensitivity of the assay?
Answer to reviewer #1: Limit of detection is determined using a dilution series of a reference sample and replicate observations. This is described for EPR in section 2.1. Clinical concordance does not assess limit of detection but rather the consistency of detection over a range of signal types and quantities when performed in the real-world diagnostic lab setting using two different methods. CAP guidelines for the clinical concordance phase of a validation study require a minimum of 20 patient cases.
- Plz check the manuscript for abbreviation as a few comes first and their full form defined in later section of the text.
Answer to reviewer #1: We have checked the manuscript for abbreviations and their full definitions. Thank you for your observation.
- Full form of few abbreviations is not given in the manuscript, plz check.
Answer to reviewer#1: Full definitions of abbreviations were included in the re-submitted version of our manuscript
- In sensitivity of EPR assay (table 1), at 0.125%VAF, the detection is 37.7% with 30ng DNA which is higher than 40ng, 50ng and 60ng of DNA. Similarly, it is happening with 0.25%VAF. Please explain it.
Answer to reviewer#1: Thank you very much for your comment. We have added the following paragraph in the Discussion, and as stated in the Conclusions, we set the validated minimum input at 40ng, and prefer 50ng for this LDT:
“In Table 1, the sensitivity at 0.125% VAF appears higher (37.7%) using 30 ng DNA input for library preparation compared to other inputs (33.3% using 40 ng, 35.5% using 50 ng and 33.3% using 60 ng) for SNVs while for indels it appears lower (14.8%) using 30 ng input compared to other inputs (19.4% using 40 ng DNA input for library preparation and 18.5% using 50 ng or 60 ng DNA input). We performed ANOVA on the replicate VAF measurements across library inputs, and 30ng input is significantly different (p 0.0012) from the others. This difference is due to greater deviation from the known reference VAF; that is, 30 ng input libraries report the variants at an average of 0.2% VAF while the others are closer to the true 0.125% VAF. We suspect 30 ng of input DNA suffers extra variability in read representation during library construction, and this is most apparent when attempting to detect sequences at very low VAFs. ANOVA for all other reference VAF standards did not show statistical significance for differences between input amounts, including 30 ng.”
Thank you very much for reviewing our manuscript.
Best regards,
Sandra V Fernandez
Reviewer 2 Report
Comments and Suggestions for Authors
Dear Authors,
Thank you for submitting your paper.
In this study, the EPR assay was assessed using reference standard DNA. In addition, a modified version of the EPR assay was created by comparison with the Tempus xF assay and applied to a sample of 29 cancer patients. The modified algorithm detected clinically relevant variants with high VAF and some of these variants showed substantially decreased VAF after treatment suggesting these are somatic. This modified version may give useful information from the patient's ctDNA regarding treatment selection, etc.
The following points should be considered.
1. P2. line 92: chr17:7579420delG: delG is a mistake for delC.
2. P2. Line 93: What is RPT?
3. P2. Lines 94-95: “TP53 variant is a deletion of C from a 5 bp homopolymer C region" is a correct sentence. C instead of G.
4. Figure 1 and 2: Please make the text in black for easier reading.
5. Please describe in the text how the 33 target genes of the EPR assay were selected.
6. The EPR assay is sensitive to detect SNVs, but not to detect indels. If the authors have any suggestions to improve this point, please discuss them.
Author Response
Reviewer #2
Dear Authors,
Thank you for submitting your paper.
In this study, the EPR assay was assessed using reference standard DNA. In addition, a modified version of the EPR assay was created by comparison with the Tempus xF assay and applied to a sample of 29 cancer patients. The modified algorithm detected clinically relevant variants with high VAF and some of these variants showed substantially decreased VAF after treatment suggesting these are somatic. This modified version may give useful information from the patient's ctDNA regarding treatment selection, etc.
Answer to reviewer#2: We would like to thank you for your time to review our manuscript. We appreciate your comments and suggestions that helped to improve our manuscript. All your comments/suggestions were included in the manuscript and are highlighted in blue.
The following points should be considered.
- line 92: chr17:7579420delG: delG is a mistake for delC.
Answer to reviewer #2: We change G by C as you mentioned. Thank you for your observation.
- Line 93: What is RPT?
Answer to reviewer #2: We had added the definition before the abbreviation. Now it is read as: were filtered as “near or within repeat region” (RPT).
- P2. Lines 94-95: “TP53 variant is a deletion of C from a 5 bp homopolymer C region" is a correct sentence. C instead of G.
Answer to reviewer #2: Thank you for pointing this out. We changed G by C as you mentioned.
- Figure 1 and 2: Please make the text in black for easier reading.
Answer to reviewer #2: Thank you for your comment. The text of Figure 1 and 2 is now in black and the size of font is bigger for easier reading. We also changed the size of the font of Figure 3.
- Please describe in the text how the 33 target genes of the EPR assay were selected.
Answer to reviewer#2: Thank you for your comment. We have added the following paragraph: “The EPR assay is a hybrid capture ctDNA-based NGS assay targeting a 33-gene panel designed to analyze genes commonly mutated in solid tumors (Suppl. Table S1A). These genes were selected by PGDx to focus on biomarkers for approved cancer therapies or relevant to established professional guidelines.”
- The EPR assay is sensitive to detect SNVs, but not to detect indels. If the authors have any suggestions to improve this point, please discuss them.
Answer to the reviewer #2: We thank the reviewer for the comment. We have added the following paragraph in the Discussion:
“EPR sensitivity for the reference indels in this study was mostly affected by post-sequencing analysis filters, which was addressed for indels with VAF ≥10% by modification of the analysis pipeline as discussed below. More generally, however, detection of indels by short-read sequencing (here 150 bp) of highly fragmented ctDNA (also approximately 150 bp in plasma) is expected to have less sensitivity com-pared to SNVs because this class of variants has a broader range of potential targets. While detection of indels less than 20 bp usually performs similarly to SNV detection, increasing indel length decreases sensitivity due to library representation and sequence alignment difficulties. Assessment of long indels will thus require complementary assays, such as analysis of intact genomic DNA from circulating tumor cells or tissue biopsies, combined with long-read sequencing or probe hybridization-based detection methods.”
Thank you very much for reviewing our manuscript.
Best regards,
Sandra V. Fernandez